# Threshold-based exploitation of noisy label in black-box unsupervised domain adaptation

**Huiwen Xu, Jaeri Lee, U Kang** *

Data Mining Lab, Seoul National University, Seoul, Republic of Korea

* ukang@snu.ac.kr

**Data availability statement:** All relevant data can be downloaded from the following

## Abstract

How can we perform unsupervised domain adaptation when transferring a black-box source model to a target domain? Black-box Unsupervised Domain Adaptation focuses on transferring the labels derived from a pre-trained black-box source model to an unlabeled target domain. The problem setting is motivated by privacy concerns associated with accessing and utilizing source data or source model parameters. Recent studies typically train the target model by mimicking the labels derived from the black-box source model, which often contain noise due to domain gaps between the source and the target. Directly exploiting such noisy labels or disregarding them may lead to a decrease in the model's performance. We propose Threshold-Based Exploitation of Noisy Predictions (TEN), a method to accurately learn the target model with noisy labels in Black-box Unsupervised Domain Adaptation. To ensure the preservation of information from the black-box source model, we employ a threshold-based approach to distinguish between *clean* labels and *noisy* labels, thereby allowing the transfer of high-confidence knowledge from both labels. We utilize a flexible thresholding approach to adjust the threshold for each class, thereby obtaining an adequate amount of clean data for hard-to-learn classes. We also exploit knowledge distillation for clean data and negative learning for noisy labels to extract high-confidence information. Extensive experiments show that TEN outperforms baselines with an accuracy improvement of up to 9.49%.

## Introduction

How can we transfer the knowledge from a black-box source model to a target task? Unsupervised domain adaptation (UDA) has emerged as a crucial research topic in the field of machine learning and computer vision. The goal of UDA is to adapt a model trained on a source domain with labeled data to a target domain with only unlabeled data, where the target domain has similar but different statistical characteristics to the source domain. The ability to perform UDA is essential in many real-world applications, such as image classification, object recognition, and natural language processing, where the target domain may not have labeled data for training.

URLs: https://faculty.cc.gatech.edu/judy/domainadapt/, https://www.hemanthdv.org/officeHomeDataset.html, https://www.imageclef.org/2014/adaptation, https://gitlab.com/tringwald/adaptiope, https://ai.bu.edu/visda-2017/ The authors do not own these datasets and had no special access privileges that others would not have.

**Funding:** This work was supported by Institute of Information & communications Technology Planning & Evaluation(IITP) grant funded by the Korea government(MSIT) [No.2022-0-00641, XVoice: Multi-Modal Voice Meta Learning], [No.RS-2020-II200894, Flexible and Efficient Model Compression Method for Various Applications and Environments], [No.RS-2021-II211343, Artificial Intelligence Graduate School Program (Seoul National University)], and [NO.RS-2021-II212068, Artificial Intelligence Innovation Hub (Artificial Intelligence Institute, Seoul National University)]. The Institute of Engineering Research at Seoul National University provided research facilities for this work. The ICT at Seoul National University provides research facilities for this study.

**Competing interests:** The authors have declared that no competing interests exist.

Unsupervised domain adaptation [1,2] has been shown to have limitations, one of which involves the necessity to access either the source data or a white-box source model. Nonetheless, sharing the source data might not be suitable due to privacy concerns, particularly in sensitive domains such as medical records or financial data. Additionally, even transferring a pretrained white-box source model to a target domain raises security concerns, as the source data could potentially be reconstructed using techniques like generative adversarial training [3].

Recent studies focus on a new problem setting known as *Black-box Unsupervised Domain Adaptation (Black-box UDA)*, where the source domain provides only a black-box model without revealing its model parameters. In this scenario, the knowledge that can be transferred to the target domain is limited to the outputs produced by the black-box source model. However, the outputs contain noise due to the intrinsic dissimilarities between the source and target domains, which makes the domain adaptation process more challenging. DINE [4] adopts knowledge distillation and pseudo-labeling strategies, instructing the target model to distill the labels produced by the black-box source model. IterLNL [5] utilizes a noisy labeling technique to select clean instances from the target data, thereby training the model solely on clean data. Both algorithms may result in decreased performance, as DINE is susceptible to learning from mislabeled data, while IterLNL experiences information loss owing to its exclusion of noisy data during training.

In this paper, we propose Threshold-Based Exploitation of Noisy Predictions (TEN), a precise method for Black-box UDA by distilling reliable high-confidence knowledge from the source labels. Owing to the presence of noise in the outputs from the source model, we partition the target data into distinct *clean* and *noisy* subsets, and apply distinct strategies to distill the high-confidence knowledge. Pseudo-labels associated with the clean subset align closely with the actual ground truths, whereas those linked to the noisy subset frequently deviate from them. In the process of data partitioning, a flexible threshold is determined for each class to ensure that hard-to-learn classes possess an adequate number of untainted instances. We harness knowledge distillation on the clean subset to emulate the source model's labels. Conversely, on the noisy subset, we exploit negative learning to discern which classes the instances do not pertain to. In addition, we employ consistency regularization coupled with entropy regularization techniques to learn the structural features of the target domain. Extensive experiments shows that TEN surpasses baseline methods, with an accuracy increase of up to 9.49%.

Our contributions are summarized as follows:

- **Algorithm.** We propose TEN, a precise method for distilling reliable high-confidence knowledge from the outputs of a black-box source model, even in the presence of noise.
- **Accuracy.** Extensive experiments conducted on real-world datasets demonstrate that TEN outperforms baselines with up to 9.49% higher accuracy for single-source UDA, and 4.81% higher accuracy for multi-source UDA.
- **Ablation Study.** We show that the performance of TEN exhibits an upward trend when more noisy labels are used for training.

Table 1 provides the definitions of symbols used in this paper.

## Related works

### Unsupervised domain adaptation

Unsupervised model adaptation, also known as source-free UDA, has garnered increasing attention due to its ability to operate without accessing the source domain, making

**Table 1. Table of symbols.**

| Symbol | Terminology | Description |
|---|---|---|
| $f_s$ | source model | black-box neural network where only network labels are available |
| $f_t$ | target model | neural network that classifies target inputs |
| $D_s$ | source data | labeled source data |
| $D_t$ | target data | unlabeled target data |
| $x_s^i$ | source input | feature of $i$-th instance in source domain |
| $y_s^i$ | source label | label of $i$-th instance in source domain |
| $x_t^i$ | target input | feature of $i$-th instance in target domain |
| $\tilde{y}_t^i$ | target pseudo label | target label of $i$-th instance from source model |
| $\hat{y}_t^i$ | target label | target label of $i$-th instance from target model |
| $\tilde{y}_{t,c}^i$ | target pseudo probability | target probability of $i$-th instance belonging to the $c$-th class obtained from source model |
| $\hat{y}_{t,c}^i$ | target probability | target probability of $i$-th instance belonging to the $c$-th class obtained from target model |
| $\hat{y}_{t,w}^i$ | target weak label | target label of $i$-th weakly augmented instance from target model |
| $\hat{y}_{t,w,c}^i$ | target weak probability | target probability of $i$-th weakly augmented instance belonging to the $c$-th class obtained from target model |
| $\hat{y}_{t,s,c}^i$ | target strong probability | target probability of $i$-th strongly augmented instance belonging to the $c$-th class obtained from target model |
| $n_s$ | source data size | number of source instances |
| $n_t$ | target data size | number of target instances |
| $\mathcal{X}_t$ | source input space | feature space in source domain |
| $\mathcal{Y}_s$ | source label space | label space in source domain |
| $\mathcal{X}_t$ | target input space | feature space in target domain |
| $\mathcal{Y}_t$ | target label space | label space in target domain |
| $c$ | class | class for source and target domains |
| $\tau_p, \tau_n$ | predefined thresholds | predefined positive and negative thresholds |
| $\mathrm{T}_p(\cdot)$ | flexible threshold | adjusted positive threshold for clean data |
| $\gamma$ | regularization threshold | constant threshold for consistency regularization |
| $\lambda_1, \lambda_2, \lambda_3$ | balancing parameters | weights for losses |
| $L_{kd}$ | knowledge distillation loss | knowledge distillation loss for clean subset |
| $L_{nce}$ | negative learning loss | negative loss for noisy subset |
| $L_{er}$ | entropy regularization loss | entropy regularization loss for all target data |
| $L_{cr}$ | consistency regularization loss | consistency regularization loss for all target data |
| $L_{total}$ | overall loss | total loss for target data |
| $L_s$ | source loss | source loss for smoothed label vectors |
| $q_s$ | source prediction | smoothed prediction in source domain |
| $\epsilon$ | smoothing parameter | parameter for label smoothing |
| $C$ | number of classes | number of classes in source and target domains |

it suitable for more practical scenarios. Early researches [6] provide a theoretical analysis of transfer learning, which motivated deep domain adaptation without source data. Zhu et al. [7] enhance domain adaptation by leveraging high-order graphs and low-rank tensors. Zhu et al. [8] propose a multiview latent space framework for UDA and MultiDA with selective pseudo-labeling. These methods are limited to solving UDA problems which does not fundamentally address privacy concerns.

In this paper, we tackle an even more challenging problem by leveraging only the predictions from a black-box model trained in the source domain for model adaptation. Few works have been conducted in this field. Zhang et al. [5] focus on selecting clean instances from noisy data, training the model only on these clean samples. However, this approach has a risk of information loss by excluding noisy data, potentially limiting the model's generalization ability. Liang et al. [4] employ a knowledge distillation and pseudo-labeling strategy, where the target model distills labels from a black-box source model. Zhang et al. [9] use a

bi-directional memorization mechanism to identify useful features and progressively correct noisy pseudo labels, improving generalization across visual recognition tasks. However, it is prone to learning from mislabeled data, which can reduce the model's performance. In contrast, our TEN approach divides the target dataset into "clean" and "noisy" subsets, and distills high-confidence predictions from both, allowing the model to leverage information from both clean and noisy data. This strategy mitigates the risks of information loss and mislabeled data, offering a more robust learning process.

## Semi-supervised learning with noisy labels

Noise can be easily accumulated during training when incorrect predictions are used in semi-supervised or unsupervised learning [10]. Such noise can cause the model to overfit to the noisy feature space, making it challenging to adapt to new domains [11]. In UDA, pseudo-labeling [12,13] and knowledge distillation [14,15] are effective techniques, but their performances can be degraded by noise. In particular, for transfer learning tasks involving distant domains, the pseudo labels for the target domain can be extremely noisy, resulting in a deterioration of subsequent training. Our proposed method in this work addresses the issue using (1) a flexible threshold technique which distills more instances for hard-to-learn classes, and (2) pseudo-labeling with negative learning which distills the information from noise.

## Proposed method

Given a black-box source model $f_s$ and unlabeled target data $D_t$, our objective is to train a target model $f_t$ that performs well on the target data without accessing any source data or source model parameters. The target data $D_t = \{x_t^i\}_{i=1}^{n_t}$ consists of $n_t$ instances distributed across $C$ categories, where $x_t^i \in \mathcal{X}_t$; $\mathcal{X}_t$ represents the target input space. The source model is pre-trained using labeled source data $D_s = \{(x_s^i, y_s^i)\}_{i=1}^{n_s}$, which contains $n_s$ labeled instances also in $C$ categories, where $x_s^i \in \mathcal{X}_s$ and $y_s^i \in \mathcal{Y}_s$. $\mathcal{X}_s$ and $\mathcal{Y}_s$ represent the source input and label spaces, respectively. We assume that the source label space $\mathcal{Y}_s$ and the target label space $\mathcal{Y}_t$ are identical, but the source and target input data have different distributions, *i.e.*, $P(\mathcal{X}_s) \neq P(\mathcal{X}_t)$. Distinctively diverging from Unsupervised Domain Adaptation, which mandates access to either the source data or its model parameters, the Black-box Unsupervised Domain Adaptation (Black-box UDA) facilitates the training of the target model in the absence of both source data or source parameters. Black-box UDA depends solely on the soft labels generated by the source model for target instances, denoted as $\tilde{y}_t^i = f_s(x_t^i)$.

### Overview

The challenge of Black-box UDA resides in distilling the knowledge from the outputs of the black-box source model. Due to the dissimilarities between the source and target domains, the outputs generated by the black-box source model contain noise. Such noise can yield erroneous results, further exacerbating the challenge of accurately labeling the target data. Consequently, it is imperative to train the target model effectively by utilizing soft labels even in the presence of such noise.

The following detailed challenges need to be addressed for the goal.

**C1** How can we effectively divide the target data into clean and noisy subsets? Utilizing a fixed high threshold for data separation may lead to extreme cases where no training data are selected for hard-to-learn classes.

**C2** How can we distill meaningful information from noisy labels? When the gap between the source and target domains is substantial, the amount of noisy labels increases, and

a failure to effectively learn from them can significantly impede the target model's performance.

**C3** How can we learn the structural information about the target data? Insufficient exploration of hidden representations leads to diminished performance of the target model owing to the disregard of the target domain's structure.

We address the aforementioned challenges with the following main ideas:

**I1 Flexible Threshold.** We design a flexible threshold for each class, thereby facilitating the allocation of a larger amount of data to those classes that are difficult to learn.

**I2 Negative Learning.** We distill the information that reflects the absence of certain classes from the noisy labels.

**I3 Structural Regularization.** We exploit entropy regularization and consistency regularization so that the target model learns intrinsic data structure about the target data.

We propose TEN, an accurate method for Black-box UDA. The overview of the proposed TEN is depicted in Fig. 1. The entire procedure comprises two distinct phases: division and training. In the division phase, given a predefined high threshold, we count the number of instances of each class whose confidences surpass the threshold, and subsequently adjust the threshold for each class based on these counts. The target data are divided into clean and noisy subsets in accordance with the adjusted thresholds. Throughout the training phase, soft labels of the target data are generated by leveraging the black-box source model. The target model mimics the soft labels of clean and noisy data via knowledge distillation and negative learning, respectively. In order to facilitate the acquisition of the structural information of the target, we exploit entropy regularization and consistency regularization. These ideas cohesively establish a comprehensive strategy for enhancing the performance of the target model by exploiting the strengths of the black-box source model and structural information in the target data.

## Flexible threshold

How can we select clean labels from the target data so that reliable knowledge can be learned during knowledge distillation? Clean subset comprises instances whose soft labels generated by the black-box source model closely align with the ground truths of the target task. Conversely, noisy subset primarily consists of instances whose soft labels tend to be inaccurate. Our goal is to train the target model to mimic only clean labels of the black-box source model, since the noisy labels may mislead the target model.

A naive technique involves a predefined high threshold to split instances into (1) clean instances whose confidences surpass the threshold, and (2) noisy instances whose confidences fall below the threshold. Noisy instances easily have wrong predictions due to the gap between the source and target domains. Thus, we consider only the soft labels of clean instances as teacher labels for knowledge distillation. Nonetheless, distinct classes have different properties for training; utilizing identical thresholds for each class to select clean instances may result in an undesirable scenario where hard-to-learn classes cannot identify appropriate instances for training the model, ultimately leading to inadequate performance.

We propose to use a flexible threshold to set a lower threshold for classes difficult to learn. When the threshold is high, the number of predictions that belong to a certain class and

## Division Phase

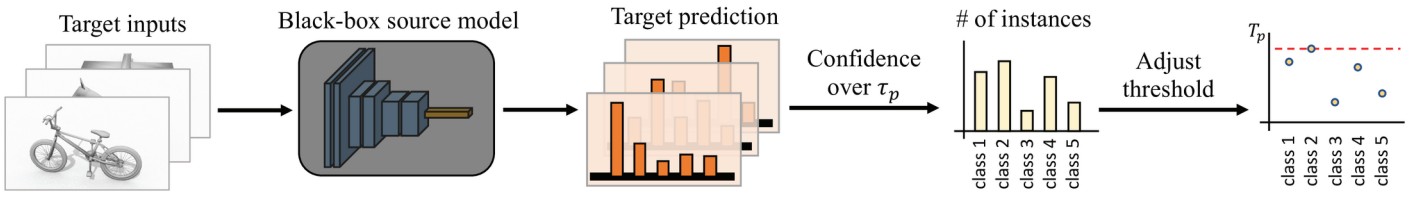

## Training Phase

**Fig 1. The overall structure of TEN.** During the division phase, we use a flexible threshold for each class by taking into account the number of instances whose confidences exceed a predefined threshold. In the training phase, soft labels of the target data are generated by leveraging the black-box source model. The target model is designed to mimic the soft labels by utilizing knowledge distillation and negative learning. Additionally, entropy regularization and consistency regularization are exploited to learn the structural information of the target.

exceed the threshold represents the learning difficulty of the class [16]. We count the number of instances whose confidence exceeds the predefined threshold and belong to the class $c$:

$$\alpha(c) = \sum_{i=1}^{n_t} \mathbb{1}\left[\max(\tilde{y}_t^i) > \tau_p\right] \cdot \mathbb{1}\left[\arg\max(\tilde{y}_t^i) = c\right] \tag{1}$$

where $\tilde{y}_t^i$ is the soft label of $i$-th target instance generated by the black-box source model, *i.e.*, $\tilde{y}_t^i = f_s(x_t^i)$, $\tau_p$ represents a predefined positive threshold, and $n_t$ represents the number of target instances. We scale the threshold for each class:

$$\mathrm{T}_p(c) = \beta(c) \cdot \tau_p \quad \text{and} \quad \beta(c) = \frac{\alpha(c)}{\max_c \alpha(c)} \tag{2}$$

where $\mathrm{T}_p(c)$ is the flexible confidence threshold for class $c$, and $\beta(c) \in [0 \sim 1]$ represents the scale factor for class $c$. The flexible threshold is computed by multiplying the predefined threshold with the scale factor, and enables the selection of a greater quantity of clean labels for training. We train the target model by mimicking the labels of the

source model on the selected clean instances whose confidence exceed the adjusted threshold:

$$L_{kd} = -\sum_{i=1}^{n_t} \mathbb{1}\big[\max(\tilde{y}_t^i) \geq \mathrm{T}_p(\arg\max(\tilde{y}_t^i))\big] \sum_{c=1}^{C} \tilde{y}_{t,c}^i \log \hat{y}_{t,c}^i \tag{3}$$

where $\hat{y}_t^i$ represents the soft label of $i$-th target instances obtained from the target model, *i.e.*, $\hat{y}_t^i = f_t(x_t^i)$. $\tilde{y}_{t,c}^i$ and $\hat{y}_{t,c}^i$ represent the probabilities of the $i$-th instance for class $c$, generated by the source and target models, respectively.

IterLNL [5] also suggests the noise rate technique to adjust class-wise threshold, but it requires many hyperparameters which are sensitive to target performance. We automatically adjust the threshold for each class according to the outputs of the black-box source model, which does not need any extra validation set or hyperparameters.

## Negative learning

The strategy of training the target model using only clean labels results in a reduction of available training data and information loss. Although the confidence of noisy labels may be too low for knowledge distillation, it does not imply that noisy labels lack learnable information. Indeed, the model's confidence in data is reflected not only in the presence of certain classes, but also in their absence [17,18]. We select a subset of the noisy labels whose confidence is low enough, and employ negative learning on them.

Given a predefined negative threshold $\tau_n$, we apply negative cross-entropy for the selected labels of noisy instances whose probability falls beneath the negative confidence threshold.

$$L_{nce} = -\sum_{i=1}^{n_t} \mathbb{1}\big[\min(\tilde{y}_t^i) \leq \tau_n\big] \frac{1}{\sum_c \mathbb{1}\big[\tilde{y}_{t,c}^i \leq \tau_n\big]} \sum_{c=1}^{C} \mathbb{1}\big[\tilde{y}_{t,c}^i \leq \tau_n\big](1 - \tilde{y}_{t,c}^i) \log(1 - \hat{y}_{t,c}^i) \tag{4}$$

where $\sum_c \mathbb{1}\big[\tilde{y}_{t,c}^i \leq \tau_n\big]$ represents the number of selected labels for the $i$-th instance.

## Structural regularization

Knowledge distillation and negative learning are designed to learn reliable high-confidence knowledge from soft labels. However, they may not capture the intrinsic data structure of the target data [4]. To address the issue, we exploit consistency regularization and entropy regularization strategies.

We assume that the target model yields similar labels when provided with diverse augmentations of the same target input [19]. The target model is trained by strongly augmented target instances, which are supervised by the pseudo-labels derived from weakly augmented target instances, as follows:

$$L_{cr} = -\sum_{i=1}^{n_t} \mathbb{1}\big[\max(\hat{y}_{t,w}^i) \geq \gamma\big] \cdot \sum_{c=1}^{C} \hat{y}_{t,w,c}^i \log \hat{y}_{t,s,c}^i \tag{5}$$

where $\hat{y}_{t,w}^i$ denotes the target labels of the $i$-th instance under weak augmentations. $\hat{y}_{t,w,c}^i$ and $\hat{y}_{t,s,c}^i$ denote the probabilities of the $i$-th instance for class $c$ under weak and strong augmentations, respectively. $\gamma$ represents a threshold to ensure that the target model is trained on high-confidence instances. We employ a flip-and-shift augmentation for the weak augmentation, while the strong augmentation is achieved by RandAugment [20].

Entropy regularization operates under the premise that the probability distribution of a well-trained model's outputs resembles a one-hot vector across classes. The target model is

trained under the supervision of the hard labels derived from the weakly augmented target instances:

$$L_{er} = -\sum_{i=1}^{n_t} \mathbb{1}\left[\max(\hat{y}_{t,w}^i) \geq \gamma\right] \sum_{c=1}^{C} \mathbb{1}\left[\arg\max(\hat{y}_{t,w}^i) = c\right] \cdot \log \hat{y}_{t,w,c}^i \tag{6}$$

Summarizing all the losses, the overall objective is formulated as:

$$L_{total} = L_{kd} + \lambda_1 L_{nce} + \lambda_2 L_{cr} + \lambda_3 L_{er} \tag{7}$$

where $\lambda_1$, $\lambda_2$, and $\lambda_3$ are balancing hyperparameters. We present the complete algorithm for TEN in Algorithm 1.

## Experiments

We present experimental results to answer the following questions about TEN:

**Q1 Classification accuracy.** Does TEN show better accuracies than baselines on benchmarks?

**Q2 Effect of flexible threshold.** Does flexible threshold improve the target performance?

**Q3 Effect of negative learning.** Does TEN exhibit desirable performance when a large amount of noisy data are used for training?

**Q4 Ablation study.** Do our ideas, such as knowledge distillation on clean subset, negative learning, and structural information, improve the target performance?

**Q5 Hyperparameter sensitivity.** Is accuracy sensitive to the positive and negative thresholds?

**Algorithm 1 THRESHOLD-BASED EXPLOITATION OF NOISY PREDICTIONS (TEN)**

**Input:** black-box source model $f_s$, unlabeled target data $D = \{x_t^i\}_{i=1}^{n_t}$, randomly initialized target model $f_t$, and predefined thresholds $\tau_p$ and $\tau_n$

**Output:** well-trained target model $f_t$

```
1: for each class do
2:    Count the number of high-confidence instances for knowledge
      distillation using Eq. (1)
3: end for
4: Calculate positive threshold T_p(c) using Eq. (2)
5: Divide the target data into clean and noisy subsets in
   accordance with T_p(c) and τ_n
6: for each epoch do
7:    for each batch do
8:       Compute knowledge distillation loss L_kd for clean subset
         using Eq. (3)
9:       Compute negative learning loss L_nce for noisy subset
         using Eq. (4)
10:       Compute consistency regularization loss L_cr and entropy
         regularization loss L_er for all data using Eq. (5) and
         Eq. (6), respectively
11:       Compute the overall loss L_total using Eq. (7), and update
         parameters of the target model f_t
12:    end for
13: end for
```

## Experimental setup

We present datasets, models, baselines, and hyperparameters for our experiments.

*Datasets.*

We use 5 image classification datasets summarized in Table 2. Office-31[1] comprises 3 domains, Amazon (A), DSLR (D), and Webcam (W), whereas Office-Home[2] [21] encompasses 4 domains, Art (A), Clipart (C), Product (P), and Real-World (R). Image-CLEF[3] dataset is composed of 4 domains, including Caltech-256 (C), ILSVRC2021 (I), PASCAL VOC2021 (P), and Bing (B). Adaptiope[4] [22] contains 3 domains, Product (P), Real life (R), and Synthetic (S), while VisDA[5] [23] comprises Synthetic (S) and Real domains (R). The imbalance ratio in a multi-class dataset indicates the ratio of the number of instances in the least prevalent (minority) class to that in the most prevalent (majority) class.

*Models.*

We use ResNet50 as source backbone and follow DINE [4] to train the source classifier. We add a fully-connected layer at the end of backbone feature extractor and train the source model $f_s$ with label smoothing technique [24]. The loss of the source model is defined as $L_s = -\mathbb{E}_{(x_s,y_s) \in \mathcal{X}_s \times \mathcal{Y}_s}(q_s)^T \log f_s(x_s)$, where $q_s = (1-\epsilon) \cdot \mathbb{1}_{y_s} + \epsilon/K$ represents the smoothed label vector. The smoothing parameter $\epsilon$ is set empirically to a value of 0.1, and $\mathbb{1}_j$ represents a one-hot encoding of $K$ dimensions where only the $j$-th value is 1. For the target model, we also use ResNet50 as backbone, and follow [2,4] to replace the original classifier with a refined architecture that consists of a bottleneck layer with 256 units and a task-specific classifier. We place a batch-normalization layer after the fully-connected layer within the bottleneck layer, and a weight normalization [25] layer in the task-specific classifier.

*Baselines.*

We compare our proposed TEN with two competitors: DINE [4] and IterLNL [5]. DINE trains the target model by distilling the soft labels derived from the black-box source model, which includes noisy data, while IterLNL focuses on training exclusively with clean data, which are selected based on the noise rate. Additionally, we train the target model using only the target predictions from the black-box source model, and the method is denoted as "No adapt."

*Hyperparameters.*

We conduct experiments five times and report the average accuracies. For all experiments, we use PyTorch on a GeForce RTX 3080. Following DINE [4], the models initialized from

**Table 2. Summary of datasets.**

| Dataset | # of instances | # of classes | Imbalance ratio |
|---|---|---|---|
| Office-31[1] | 4,110 | 31 | 22.58% |
| Office-Home[2] | 15,588 | 65 | 15.15% |
| Image-CLEF[3] | 2,400 | 12 | 100% |
| Adaptiope[4] | 36,900 | 123 | 100% |
| VisDA[5] | > 280,000 | 212 | 34.19% |

[1] https://faculty.cc.gatech.edu/ judy/domainadapt/
[2] https://www.hemanthdv.org/officeHomeDataset.html
[3] https://www.imageclef.org/2014/adaptation
[4] https://gitlab.com/tringwald/adaptiope
[5] https://ai.bu.edu/visda-2017/

the pre-trained ImageNet model have their learning rate set to 1e-3, and those learned from scratch are set to a learning rate of 1e-2. Furthermore, we adopt learning rate scheduler, momentum (0.9), weight decay (1e-3), bottleneck size (256), and batch size (64). The values for the thresholds $\tau_p$ and $\tau_n$ are determined to be within the sets {0.5, 0.6, 0.7, 0.8} and {0.0001, 0.0005, 0.001, 0.005, 0.01}, respectively. The hyperparameters are optimized using Optuna.

## Classification accuracy (Q1)

We report the target accuracies on the five datasets in Tables 3∼7. TEN achieves the highest average accuracy in most cases and surpasses the second-best method by up to 7.59%, 4.08%, 3.67%, 9.49%, and 3.62% for Office-31, Office-Home, Image-CLEF, Adaptiope, and VisDA, respectively. Despite the significant disparity between the synthetic domain (Synthetic) and real-world (Product or Real life) domains in Adaptiope, TEN exhibits considerable improvement which shows that TEN is an effective method for domain adaptation.

We also evaluate the performance of Black-box UDA with multiple source models. The soft labels of the source models are aggregated through averaging to establish the initialized soft label of the source models. We conduct experiments on four multi-source datasets, and use ResNet101 as backbone. As shown in Table 9, TEN demonstrates competitive performance across all these datasets, surpassing competitors by up to 4.81%.

## Effect of flexible threshold (Q2)

To demonstrate the effectiveness of the flexible threshold, we compare the target performance against various division strategies as presented in Table 8. "No div." skips the division phase,

**Table 3. Accuracies (%) on Office-Home for black-box model adaptation. The best is in bold. TEN outperforms baselines with up to 4.08% higher accuracy.**

| Method | A→C | A→P | A→R | C→A | C→P | C→R | P→A | P→C | P→R | R→A | R→C | R→P |
|---|---|---|---|---|---|---|---|---|---|---|---|---|
| No adapt. | 44.24 | 66.68 | 74.20 | 53.23 | 62.65 | 64.82 | 52.90 | 40.96 | 73.61 | 66.38 | 47.06 | 77.20 |
| DINE | 52.88 | 78.01 | **81.91** | 64.44 | 75.72 | 78.82 | 62.34 | 49.28 | **82.26** | 71.03 | 55.95 | 84.25 |
| IterLNL | 50.47 | 78.48 | 78.34 | 65.95 | 77.15 | 75.92 | 61.84 | 49.01 | 78.09 | 67.58 | 53.56 | 84.58 |
| TEN (proposed) | **55.01** | **78.55** | 81.29 | **66.79** | **78.89** | 79.37 | **66.42** | **51.75** | 81.75 | **72.31** | **58.49** | **85.42** |

**Table 4. Accuracies (%) on Image-CLEF for black-box model adaptation. The best is in bold. TEN outperforms baselines with up to 3.67% higher accuracy.**

| Method | B→C | B→I | B→P | C→B | C→I | C→P | I→B | I→C | I→P | P→B | P→C | P→I |
|---|---|---|---|---|---|---|---|---|---|---|---|---|
| No adapt. | 89.00 | 83.67 | 67.17 | 60.17 | 83.50 | 71.83 | 61.50 | 92.33 | 76.00 | 60.00 | 90.83 | 89.00 |
| DINE | 96.00 | 93.67 | 78.17 | 64.83 | 91.50 | 77.83 | 64.17 | 96.67 | 79.00 | 63.83 | 96.00 | 93.17 |
| IterLNL | 94.67 | 93.17 | 76.83 | 65.50 | 91.17 | 78.50 | 64.67 | 95.00 | 78.67 | 64.00 | 95.17 | 93.50 |
| TEN (proposed) | **97.00** | **94.83** | **80.00** | **66.33** | **93.67** | **80.50** | **66.50** | **97.67** | **79.67** | **67.67** | **97.50** | **95.00** |

**Table 5. Accuracies (%) on Office-31 for black-box model adaptation. The best is in bold. TEN outperforms baselines with up to 7.59% higher accuracy.**

| Method | A→D | A→W | D→A | D→W | W→A | W→D |
|---|---|---|---|---|---|---|
| No adapt. | 80.12 | 76.98 | 57.15 | 92.70 | 61.02 | 98.39 |
| DINE | 94.18 | 86.67 | 71.67 | 93.71 | 73.09 | 99.20 |
| IterLNL | 94.02 | 87.13 | 71.49 | 92.96 | 69.40 | 99.28 |
| TEN (proposed) | **95.38** | **94.72** | **74.83** | **98.24** | **76.22** | **99.80** |

**Table 6. Accuracies (%) on Adaptiope for black-box model adaptation. The best is in bold. TEN outperforms baselines with up to 9.49% higher accuracy.**

| Method | P→R | P→S | R→P | R→S | S→P | S→R |
|---|---|---|---|---|---|---|
| No adapt. | 67.03 | 33.62 | 87.54 | 29.61 | 10.28 | 2.10 |
| DINE | **78.85** | 44.55 | 91.71 | 40.69 | 18.50 | 4.84 |
| IterLNL | 76.36 | 48.76 | 88.64 | 41.79 | 15.97 | 4.07 |
| TEN (proposed) | 78.54 | **53.33** | **91.80** | **50.37** | **25.46** | **6.09** |

**Table 7. Accuracies (%) on VisDA for black-box model adaptation. The best is in bold. TEN outperforms baselines with up to 3.62% higher accuracy.**

| Method | S→R | R→S |
|---|---|---|
| No adapt. | 39.92 | 58.84 |
| DINE | 60.95 | 76.61 |
| IterLNL | 62.27 | 73.21 |
| TEN (proposed) | **65.73** | **80.23** |

**Table 8. Accuracies (%) on Office-31 for various division strategies used for splitting the target data into clean and noisy subsets. The best is in bold. Our proposed TEN outperforms baselines, demonstrating that flexible threshold is effective by taking advantages of high-confidence clean labels.**

| Method | A→D | A→W | D→A | D→W | W→A | W→D |
|---|---|---|---|---|---|---|
| No div. | 94.62 | 96.77 | 74.19 | 97.85 | 75.27 | 97.85 |
| TEN (fixed) | 83.87 | 77.42 | 72.04 | 96.77 | 73.12 | 97.85 |
| TEN (proposed) | **95.38** | **94.72** | **74.83** | **98.24** | **76.22** | **99.80** |

**Table 9. Accuracies (%) on four datasets for multi-source model adaptation. The best is in bold. TEN outperforms baselines with up to 4.81% higher accuracy.**

| Method | Office-Home | | | | Image-CLEF | | | | Office-31 | | | Adaptiope | | |
|---|---|---|---|---|---|---|---|---|---|---|---|---|---|---|
| | →A | →C | →P | →R | →B | →C | →I | →P | →A | →D | →W | →P | →R | →S |
| No Adapt. | 54.98 | 49.90 | 69.64 | 76.77 | 61.64 | 92.18 | 87.47 | 72.40 | 64.55 | 82.34 | 80.75 | 73.96 | 63.90 | 33.15 |
| DINE | 74.95 | 64.15 | 84.63 | 84.79 | 64.99 | 97.87 | 93.08 | 79.79 | 77.19 | 99.21 | 98.28 | 73.48 | 73.60 | 45.33 |
| IterLNL | **76.29** | **65.19** | 82.48 | 80.50 | 63.77 | 97.17 | 94.19 | 80.09 | 77.06 | 97.38 | 96.38 | 78.84 | 72.98 | 45.53 |
| TEN (proposed) | 76.09 | 65.41 | **86.74** | **86.08** | **66.29** | **98.32** | **94.51** | **81.47** | **79.27** | **99.27** | **98.76** | **83.65** | **75.25** | **46.25** |

and leverages all soft labels including noise for knowledge distillation. "TEN (fixed)" chooses clean labels based on a predefined threshold to ensure that the labels possess high confidence. Notably, our proposed TEN surpasses the baselines, demonstrating that the flexible threshold is beneficial by fully harnessing clean labels with high-confidence. We analyze the reason by plotting the instances (clean subset) utilized for knowledge distillation in Fig. 2. "Pos" and "Neg" represent instances that are labeled correctly and inaccurately, respectively. With a lower fixed threshold as in Fig. 2 (a), all instances including noise are incorporated into training, which potentially undermine the target performance. In contrast, a higher fixed threshold ensures the selection of high-confidence instances but might also curtail the overall number of "Pos" training instances as shown in Fig. 2 (b). Compared to the baselines, TEN proposes a flexible threshold to diminish the number of "Neg" instances in the selection process while simultaneously ensuring the inclusion of a sufficient quantity of "Pos" instances as shown in Fig. 2 (c). This closely aligns with the ideal scenario where all "Pos" instances are employed for distillation, while precluding the inclusion of all "Neg" instances.

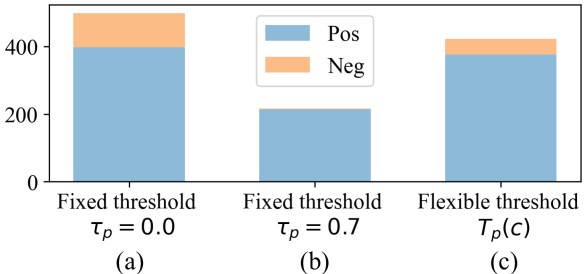

**Fig 2. Impact of threshold on instance selection.** The stacked bars indicate the number of instances used for knowledge distillation in each case. "Pos" and "Neg" represent instances that are labeled correctly and inaccurately, respectively. (a) When the fixed threshold is low, all instances, including noise, are used for training, which may diminish the target performance. (b) Conversely, when the fixed threshold is high, high-confidence instances are selected, but with the disadvantage of reducing a significant number of training instances. (c) TEN mitigates the occurrence of "Neg" instances in the selection process, while concurrently ensuring the inclusion of an adequate number of "Pos" instances. It closely approximates the optimal scenario where all positive instances are utilized for distillation, while excluding all negative instances.

## Effect of negative learning (Q3)

We conduct experiments on Office-31 dataset with varying quantities of noisy instances for training. In Table 10, the percentage in the first column indicates the proportion of noisy data used for training relative to the total amount of noisy data available. We have two observations. First, TEN achieves the highest accuracy with an improvement of up to 7.59% in all cases. Second, the relative performance of TEN compared to competitors increase as more noisy data are used training. The reason is that TEN effectively extracts high-confidence information from the noisy data, mitigating information loss.

## Ablation study (Q4)

We examine the contribution of various components of TEN in Table 11. The proposed flexible threshold, negative learning, entropy regularization, and consistency regularization loss

**Table 10. Accuracies (%) on Office-31 for different number of noisy instances used for negative learning. The best is in bold. The percentage in the first column indicates the proportion of noisy data used for training relative to the entire noisy data. Note that TEN outperforms competitors in almost all the cases. Also note that the performance gap of TEN and competitors increases as more noisy data are used for training.**

| Noisy data for training | Method | A→D | A→W | D→A | D→W | W→A | W→D |
|---|---|---|---|---|---|---|---|
| 100% | No adapt. | 80.12 | 76.98 | 57.15 | 92.70 | 61.02 | 98.39 |
| | DINE | 94.18 | 86.67 | 71.67 | 93.71 | 73.09 | 99.20 |
| | IterLNL | 94.02 | 87.13 | 71.49 | 92.96 | 69.40 | 99.28 |
| | TEN (proposed) | **95.38** | **94.72** | **74.83** | **98.24** | **76.22** | **99.80** |
| 66% | No adapt. | 78.12 | 74.82 | 55.37 | 91.63 | 58.37 | 96.64 |
| | DINE | 93.17 | 85.29 | 69.74 | 91.73 | 70.19 | 97.90 |
| | IterLNL | 91.99 | 84.74 | 70.45 | 90.22 | 67.65 | 96.94 |
| | TEN (proposed) | **92.06** | **86.87** | **72.27** | **92.89** | **72.93** | **98.20** |
| 33% | No adapt. | 75.39 | 73.00 | 53.31 | 87.39 | 55.94 | 95.64 |
| | DINE | 91.46 | 82.73 | 68.05 | **89.36** | 68.29 | 95.05 |
| | IterLNL | 90.39 | 82.30 | 68.45 | 87.75 | 65.04 | 95.85 |
| | TEN (proposed) | **91.82** | **83.95** | **69.57** | 88.09 | **68.58** | **96.06** |

**Table 11. Ablation study for TEN. The best is in bold. TEN achieves the highest accuracy among its variants, demonstrating that the main ideas of TEN are effective for its superior performance.**

| $L_{kd}$ | $L_{nce}$ | $L_{er}$ | $L_{cr}$ | A→D | A→W | D→A | D→W | W→A | W→D |
|---|---|---|---|---|---|---|---|---|---|
| o | o | o |  | 91.00 | 90.53 | 70.06 | 93.95 | 73.16 | 97.39 |
| o | o |  | o | 92.35 | 91.28 | 70.90 | 94.42 | 72.67 | 96.56 |
| o |  | o | o | 93.49 | 92.82 | 73.52 | 96.80 | 75.21 | 98.38 |
|  | o | o | o | 76.89 | 75.21 | 56.49 | 76.96 | 56.95 | 77.98 |
| o | o | o | o | **95.38** | **94.72** | **74.83** | **98.24** | **76.22** | **99.80** |

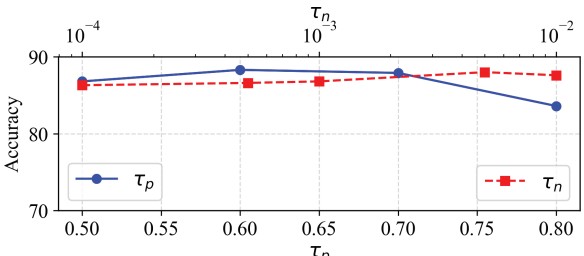

**Fig 3. Hyperparameter sensitivity to accuracy.** $\tau_p$ has a significant impact on accuracy, while $\tau_n$ demonstrates robustness with minimal effect on accuracy.

consistently enhance the accuracy of TEN with up to 21.28%, 1.90%, 3.93%, and 4.77%, respectively.

## Hyperparameter sensitivity (Q5)

Fig 3 evaluates the sensitivity of the two hyperparameters $\tau_p$ (positive threshold) and $\tau_n$ (negative threshold) on accuracy using the Office-31 dataset. The horizontal axis at the bottom represents $\tau_p$ ranging from 0.50 to 0.80, while the top horizontal axis corresponds to $\tau_n$, presented on a logarithmic scale from $10^{-4}$ to $10^{-2}$. The vertical axis indicates accuracy. For $\tau_p$, the accuracy is optimal at 0.6, while it drops when $\tau_p$ is too high because a higher $\tau_p$ reduces the number of clean samples used to adjust the threshold, leading to an inaccurate evaluation of the flexible thresholds. For $\tau_n$, the accuracy is optimal at 0.005 and remains stable across different values with only minor variations.

## Conclusion

We propose TEN, an accurate method for Black-box Unsupervised Domain Adaptation. TEN partitions the target data into clean and noisy subsets. The pseudo labels of the clean subset correspond closely to the ground truths of the target task, while those of the noisy subset are often inaccurate. The high-confidence of clean subset reflects the presence of certain classes, while the high-confidence of noisy subset reflects their absence. Considering this, we exploit knowledge distillation on clean labels and negative learning on noisy labels to learn their respective high-confidence predictions. Experimental results demonstrate that TEN outperforms the baseline methods by up to 9.49% higher accuracy for single-source UDA, and 4.81% higher accuracy for multi-source UDA. The performance of our approach exhibits an upward trend as an increasing amount of noisy data is utilized for training.

There are several possible future research directions. Our primary contribution in this work is the significant boost in accuracy. However, achieving computational efficiency is

another important aspect. Also, adapting our method for non-image data by considering their unique characteristics would be interesting. Finally, addressing the case when we have *few* unlabeled target data is a promising direction.

## Author contributions

**Conceptualization:** Huiwen Xu.

**Data curation:** Huiwen Xu.

**Formal analysis:** Huiwen Xu, U Kang.

**Funding acquisition:** U Kang.

**Investigation:** Huiwen Xu.

**Methodology:** Huiwen Xu.

**Project administration:** U Kang.

**Resources:** Huiwen Xu, U Kang.

**Software:** Huiwen Xu.

**Supervision:** U Kang.

**Validation:** Huiwen Xu, Jaeri Lee.

**Visualization:** Huiwen Xu.

**Writing – original draft:** Huiwen Xu.

**Writing – review & editing:** Huiwen Xu, Jaeri Lee, U Kang.

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
