## [Decision Letter · Decision Letter 0]

26 Nov 2024

PONE-D-24-42138Threshold-Based Exploitation of Noisy Label in Black-box Unsupervised Domain AdaptationPLOS ONE

Dear Dr. Kang,

Thank you for submitting your manuscript to PLOS ONE. After careful consideration, we feel that it has merit but does not fully meet PLOS ONE’s publication criteria as it currently stands. Therefore, we invite you to submit a revised version of the manuscript that addresses the points raised during the review process.

We look forward to receiving your revised manuscript.

Kind regards,

Lei Chu

Academic Editor

PLOS ONE

Journal Requirements:

2. Please note that PLOS ONE has specific guidelines on code sharing for submissions in which author-generated code underpins the findings in the manuscript. In these cases, we expect all author-generated code to be made available without restrictions upon publication of the work. 

Please review our guidelines at https://journals.plos.org/plosone/s/materials-and-software-sharing#loc-sharing-code and ensure that your code is shared in a way that follows best practice and facilitates reproducibility and reuse.

“This work was supported by Institute of Information \& communications Technology Planning \& Evaluation(IITP) grant funded by the Korea government(MSIT) [No.2022-0-00641, XVoice: Multi-Modal Voice Meta Learning], [No.RS-2020-II200894, Flexible and Efficient Model Compression Method for Various Applications and Environments], [No.RS-2021-II211343, Artificial Intelligence Graduate School Program (Seoul National University)], and [NO.RS-2021-II212068, Artificial Intelligence Innovation Hub (Artificial Intelligence Institute, Seoul National University)]. The Institute of Engineering Research at Seoul National University provided research facilities for this work. The ICT at Seoul National University provides research facilities for this study.”

5. We note that your Data Availability Statement is currently as follows: 

“All relevant data are within the manuscript and its Supporting Information files.”

**Additional Editor Comments:**

The study shows promise, but certain aspects require further elaboration and clarification to enhance its rigor and impact. For example, the reviewers highlight the need for a more detailed analysis of hyperparameter sensitivity, particularly thresholds and regularization parameters critical to the method's performance. They recommend including explicit comparisons of computational efficiency against baseline methods to emphasize the practical benefits. Additionally, a discussion on potential failure cases and the method's generalizability to domains beyond image classification is necessary to clarify its broader applicability and limitations.

Reviewers' comments:

Reviewer's Responses to Questions

**Comments to the Author**

1. Is the manuscript technically sound, and do the data support the conclusions?

Reviewer #1: Yes

Reviewer #2: Yes

2. Has the statistical analysis been performed appropriately and rigorously? 

Reviewer #1: Yes

Reviewer #2: Yes

3. Have the authors made all data underlying the findings in their manuscript fully available?

Reviewer #1: Yes

Reviewer #2: Yes

4. Is the manuscript presented in an intelligible fashion and written in standard English?

Reviewer #1: Yes

Reviewer #2: Yes

5. Review Comments to the Author

Reviewer #1: The authors proposed "Threshold-Based Exploitation of Noisy Label in Black-box Unsupervised Domain Adaptation". The structure of the article is well structured. But authors should follow the following comments.

1.Proofread the entire manuscript.

2.Draw a graphical abstract of your proposed approach.

3.Compare your approach with previous approaches.

4. Explain about features you are using in this study.

Reviewer #2: This paper tackles the significant challenge of Black-box Unsupervised Domain Adaptation (UDA), where source data and model parameters are inaccessible due to privacy constraints. The authors propose Threshold-Based Exploitation of Noisy Predictions (TEN), a method designed to adapt target models using noisy labels generated by a black-box source model. TEN introduces a flexible thresholding mechanism to classify data into clean and noisy subsets, effectively handling class imbalance and improving learning from high-confidence instances. It further employs knowledge distillation for clean labels, negative learning for noisy labels, and structural regularization techniques to enhance the adaptation process. Extensive experiments demonstrate that TEN achieves up to 9.49% higher accuracy than existing baselines, highlighting its robustness and practicality. The flexible threshold approach is particularly notable, addressing the imbalance and difficulty of learning from noisy labels effectively. Additionally, the integration of knowledge distillation, negative learning, and entropy regularization creates a well-rounded and efficient framework for improving target model performance. The authors support their claims with extensive experiments across multiple datasets and scenarios, including single-source and multi-source UDA, demonstrating consistent accuracy improvements. The inclusion of ablation studies further validates the significance of each component in the proposed method.

However, there are still some issues to be addressed before the paper is accepted:

1. There should be a deeper discussion of hyperparameter sensitivity, particularly the thresholds and regularization parameters critical to TEN's performance.

2. While the computational efficiency of the method is implied, explicit comparisons of overhead against baselines would strengthen the practical appeal.

3. The authors should discuss the potential failure cases or generalizability to domains beyond image classification limits the broader applicability of the findings.

4. The authors should include references related to UDA and black-box UDA in related work, such as [1-3].

[1] Zhang, J., Huang, J., Jiang, X., & Lu, S. (2023). Black-box unsupervised domain adaptation with bi-directional atkinson-shiffrin memory. In Proceedings of the IEEE/CVF International Conference on Computer Vision (pp. 11771-11782).

[2] Zhu, C., Wang, Q., Xie, Y., & Xu, S. (2024). Multiview latent space learning with progressively fine-tuned deep features for unsupervised domain adaptation. Information Sciences, 662, 120223.

[3] Zhu, C., Zhang, L., Luo, W., Jiang, G., & Wang, Q. (2024). Tensorial multiview low-rank high-order graph learning for context-enhanced domain adaptation. Neural Networks, 106859.

6. PLOS authors have the option to publish the peer review history of their article (what does this mean?). If published, this will include your full peer review and any attached files.

Reviewer #1: No

Reviewer #2: No

---

## [Author Response · Author response to Decision Letter 1]

23 Feb 2025

We have carefully addressed all your comments and concerns in the “Response to Reviewers” file.

---

## [Decision Letter · Decision Letter 1]

14 Mar 2025

Threshold-Based Exploitation of Noisy Label in Black-box Unsupervised Domain Adaptation

PONE-D-24-42138R1

Dear Dr. Kang,

We’re pleased to inform you that your manuscript has been judged scientifically suitable for publication and will be formally accepted for publication once it meets all outstanding technical requirements.

Kind regards,

Lei Chu

Academic Editor

PLOS ONE

Additional Editor Comments (optional):

Reviewers' comments:

Reviewer's Responses to Questions

**Comments to the Author**

1. If the authors have adequately addressed your comments raised in a previous round of review and you feel that this manuscript is now acceptable for publication, you may indicate that here to bypass the “Comments to the Author” section, enter your conflict of interest statement in the “Confidential to Editor” section, and submit your "Accept" recommendation.

Reviewer #2: All comments have been addressed

2. Is the manuscript technically sound, and do the data support the conclusions?

Reviewer #2: Yes

3. Has the statistical analysis been performed appropriately and rigorously? 

Reviewer #2: Yes

4. Have the authors made all data underlying the findings in their manuscript fully available?

Reviewer #2: Yes

5. Is the manuscript presented in an intelligible fashion and written in standard English?

Reviewer #2: Yes

6. Review Comments to the Author

Reviewer #2: The authors have addressed my previous concerns. The manuscript is suggested to be accepted under current version.

7. PLOS authors have the option to publish the peer review history of their article (what does this mean?). If published, this will include your full peer review and any attached files.

Reviewer #2: No

---

## [Editor Report · Acceptance letter]

PONE-D-24-42138R1

PLOS ONE

Dear Dr. Kang,

I'm pleased to inform you that your manuscript has been deemed suitable for publication in PLOS ONE. Congratulations! Your manuscript is now being handed over to our production team.

Kind regards,

on behalf of

Dr. Lei Chu

Academic Editor

PLOS ONE